# Genome-scale metabolic network reconstruction analysis identifies bacterial vaginosis-associated metabolic interactions

Lillian R. Dillard[1,4], Emma M. Glass [2,4], Glynis L. Kolling [2,4], Krystal Thomas-White[3], Fiorella Wever[3], Robert Markowitz[3], David Lyttle[3] & Jason A. Papin [1,2] ✉

Bacterial vaginosis (BV) is the most prevalent vaginal condition among reproductive-age women presenting with vaginal complaints. Despite its significant impact on women's health, limited knowledge exists regarding the microbial community composition and metabolic interactions associated with BV. In this study, we analyze metagenomic data obtained from human vaginal swabs to generate in silico predictions of BV-associated bacterial metabolic interactions via genome-scale metabolic network reconstructions (GENREs). While most efforts to characterize symptomatic BV (and thus guide therapeutic intervention by identifying responders and non-responders to treatment) are based on genomic profiling, our in silico simulations reveal functional metabolic relatedness between species as quite distinct from genetic relatedness. We grow several of the most common co-occurring bacteria (*Prevotella amnii, Prevotella buccalis, Hoylesella timonensis, Lactobacillus iners, Fannyhessea vaginae*, and *Aerrococcus christenssii*) on the spent media of *Gardnerella* species and perform metabolomics to identify potential mechanisms of metabolic interaction. Through these analyses, we identify BV-associated bacteria that produce caffeate, a compound implicated in estrogen receptor binding, when grown in the spent media of other BV-associated bacteria. These findings underscore the complex and diverse nature of BV-associated bacterial community structures and several of these mechanisms are of potential significance in understanding host-microbiome relationships.

Bacterial vaginosis (BV) is the most common state of vaginal dysbiosis among reproductive age women with vaginal complaints[1]. BV is a polymicrobial condition characterized by low levels of *Lactobacillus*, high levels of diverse anaerobes, a vaginal pH greater than 4.5, thin vaginal discharge, and a fishy odor[2]. Various factors, such as sexual activity, menstruation, antibiotics, and douching can alter a healthy, acidic, *Lactobacillus*-dominant, vaginal microbiome, and lead to the

development of BV[3-7]. BV is disproportionally prevalent among women of color, affecting 33–64% of Black women, and 31–32% of Hispanic women, compared to 23–35% of White women[8-11]. BV increases risk of contracting sexually transmitted diseases, including HIV, and is also associated with risk of preterm birth[12-14]. BV accounts for an estimated cost of $14.4 billion USD annually when considering both BV treatment costs and BV-associated healthcare costs in the United States alone[15].

[1]Department of Biochemistry & Molecular Genetics, University of Virginia, Charlottesville, VA, USA. [2]Department of Biomedical Engineering, University of Virginia, Charlottesville, VA, USA. [3]Evvy, New York, NY, USA. [4]These authors contributed equally: Lillian R. Dillard, Emma M. Glass, Glynis L. Kolling. ✉e-mail: papin@virginia.edu

Despite the significant impact of BV on women's health, there is a dearth of information regarding the microbial community composition and metabolic interactions associated with this condition[16]. By investigating the competitive and mutualistic relationships among BV-associated bacteria, we can identify potential targets for therapeutic interventions. The inclusive list of bacteria involved in BV has yet to be defined; however, certain bacterial species are commonly associated with this condition, including: *Gardnerella* species, *Prevotella bivia, Prevotella amnii, Prevotella buccalis, Hoylesella timonensis, Lactobacillus iners, Fannyhessea vaginae*, and *Aerrococcus christenssii* (Fig. 1)[17–19]. Although the *Gardnerella* genus is the primary contributor to BV, the large number of distinct *Gardnerella* species associated with BV make experimental analysis challenging[20–23]. Computational modeling allows us to simulate thousands of pairwise interactions between bacterial species to analyze possible competitive and mutualistic behaviors.

In this study, we conducted an analysis of metagenomic data obtained from human vaginal swabs (Methods) in order to ascertain the microbial composition of both symptomatic and asymptomatic cases of molecular-BV (hereafter referred to as BV)[24]. With these data, we generated in silico predictions of BV-associated bacterial metabolic interactions using genome-scale metabolic network reconstructions (GENREs). To validate these predictions, we carried out in vitro growth experiments. Furthermore, we collected bacterial supernatants to identify metabolites that putatively underlie BV-associated bacterial interactions. We subsequently conducted follow-up experiments to confirm the role of metabolites that were implicated both in silico and in vitro in contributing to competitive interactions. By elucidating the underlying mutualistic and competitive interactions between bacterial pairs, we enhance our understanding of the intricate microbial community associated with BV, ultimately leading to the development of more rigorous typing of BV and possible effective treatments for this prevalent condition.

## Results

### BV-associated symptom status shapes distinct in vivo co-occurrence patterns among bacterial species

*L. iners, G. vaginalis, F. vaginae*, and *H. timonensis* are frequently found to co-occur in both symptomatic and asymptomatic BV-associated

samples (Fig. 1). Symptomatic BV-associated samples are defined as the presence of self-reported excessive discharge and vulvovaginal pruritus. In the absence of microscopy based diagnostics, Amsel Criteria, and Nugent score, BV-associated profiles were defined by *Gardnerella* dominance via metagenomic sequencing (further details in Methods). *L. iners* was observed to co-occur with *G. vaginalis* in 27% of healthy samples, whereas the co-occurrence of *F. vaginae* and *H. timonensis* was only 4% and 3%, respectively. *A. christensenii* and *P. bivia* were identified as more common co-occurring species in symptomatic samples. On the other hand, *P. amnii, P. buccalis*, and *H. timonensis* were more commonly observed as co-occurring species with *G. vaginalis* in asymptomatic samples. Patients with asymptomatic BV-associated profiles tended to be older, but otherwise the demographic features collected from sample metadata did not appear to correlate with differences in symptomatic, asymptomatic, and healthy samples (Table 1).

Average nucleotide identity (ANI) of *Gardnerella* genomes identified 13 genetically distinct species, all of which have been previously characterized, and are defined here as 95% genetic similarity (S1). Four species are already named (*G. vaginalis, G. piotii, G. swidsinkii*, and *G. leopoldii*) with the remainder unnamed and therefore given a number (species 1-9). Within these species there were also distinct genetic clades for both *G. vaginalis* and *G. piotii* designated as clades A and B in each (S1).

### There are complex mutualistic and competitive relationships between BV-associated bacteria

We analyzed mutualism, competition, and net (sum of positive mutualism flux with negative competition flux) interactions at the single bacterial species, single interaction level. In the context of these in silico simulations, primary bacteria are the focal bacterial species whose biomass flux changes are measured and analyzed. Co-occurring bacteria are those that interact with the primary bacteria within the simulation environment. Their presence influences the biomass flux of the primary bacteria through competitive or mutualistic interactions, thereby affecting the primary bacteria's metabolic dynamics and overall community structure. Through our high-throughput pairwise simulations, each bacterial species serves as both the primary and co-occurring bacteria at different points. This means that for each interaction pair, we generate two outputs: one where bacteria 1 is the primary and bacteria 2 are co-occurring, and one where bacteria 1 are co-occurring and bacteria 2 are primary. Our investigation revealed a consistent lack of mutualism benefits across two *Prevotella* species and *H. timonensis*, which cluster closely at the primary bacteria level (Fig. 2A). In contrast, *A. christensenii* and *L. iners* showed significant mutualistic benefits in pairwise simulations. In terms of competition, a small subset of *Gardnerella* strains at the bottom-most of the heatmap were repeatedly outcompeted, as evidenced by high biomass flux decrease across almost all interactions (Fig. 2B). At the net flux level, *L. iners* and *A. christensenii* at the primary bacteria level flux values indicated consistent biomass benefit from mutualistic interactions and low biomass cost due to competition (Fig. 2C). Finally, *L. iners*, all *Prevotella* species, *H. timonensis, F. vaginae, A. christensenii*, and some *Gardnerella* strains played significant mutualistic roles in approximately half of the primary bacterial strain interactions, while playing a more neutral to competitive role in the other half of primary bacterial strain interactions.

We assessed single bacterial species across interactions—mutualistic, competitive, and net—to examine high-level patterns of bacterial similarity. We found inter-*Gardnerella* clade clustering during mutualism, including *G. leopoldii, G. swidsinksii, G. 1,2,3,4*, and unknown strains (Fig. 2D). Additionally, we observed clustering across *G. vaginalis, G. piotii, G. 5,6,7,8,9*, and *P. amnii*. Specifically, *P. amnii* clustered independently from *P. bivia* and *P. buccalis*. Conversely, there was less prevalent local structure observed at the competition level including

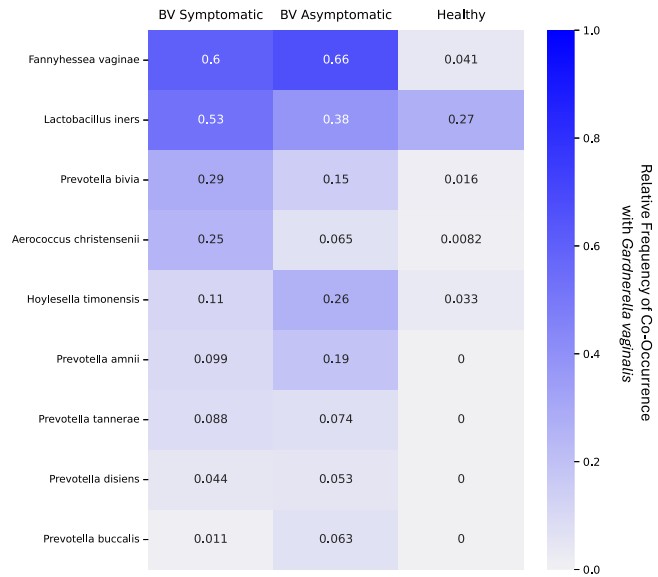

**Fig. 1 | Differential community composition of symptomatic vs asymptomatic BV.** A heatmap illustrating relative frequency of *Gardnerella vaginalis* co-occurrence with non-*Gardnerella* species in BV symptomatic, BV asymptomatic, and healthy samples.

**Table 1 | Demographics and clinical characteristics of vaginal metagenomic samples**

| Demographics and clinical characteristics | | | | |
|---|---|---|---|---|
| | **Symptomatic BV** | **Asymptomatic BV** | **Healthy** | **P-value** |
| Number of Tests | 91 | 489 | 122 | |
| Age (years) Mean (st dev) | 36.26 (7.51) | 40.33 (9.75) | 39.51 (9.9) | 0.00029[a] |
| Body Mass Index mean (st dev) | 24.34 (5.35) | 24.95 (5.56) | 23.93 (4.57) | 0.2841[a] |
| **Race/Ethnicity (Number)** | | | | **0.1015[b]** |
| White | 62 | 336 | 95 | |
| Black or African American | 15 | 68 | 11 | |
| Hispanic or Latino | 16 | 57 | 13 | |
| Asian | 5 | 40 | 2 | |
| American Indian or Alaskan Native | 0 | 7 | 6 | |
| Southeast Asian | 2 | 5 | 2 | |
| Middle Eastern | 0 | 8 | 1 | |
| South Asian | 1 | 5 | 2 | |
| Native Hawaiian | 0 | 2 | 0 | |
| Other | 0 | 7 | 2 | |
| Unknown | 0 | 0 | 0 | |
| Prefer not to say | 0 | 9 | 3 | |
| **Which of these describes your sexual activity? (Number)[a]multiselect** | | | | **0.7116[b]** |
| One partner | 64 | 335 | 86 | |
| Penetrative vaginal sex | 58 | 284 | 69 | |
| Male partner(s)[a]assigned at birth | 47 | 216 | 58 | |
| Receiving oral sex | 48 | 216 | 45 | |
| Sex toys | 33 | 189 | 44 | |
| Multiple partners | 8 | 51 | 12 | |
| Receiving anal sex | 8 | 43 | 13 | |
| Female partner(s)[a]assigned at birth | 8 | 19 | 2 | |
| Prefer not to say | 0 | 6 | 0 | |
| **When were you last sexually active? (Number)** | | | | **0.3803[b]** |
| Within the past 5 days | 11 | 143 | 40 | |
| Within the past 2 weeks | 14 | 73 | 15 | |
| Within the past 30 days | 7 | 46 | 11 | |
| Previously sexually active (over a month ago) | 8 | 68 | 19 | |
| Never sexually active | 0 | 7 | 0 | |
| Prefer not to say | 0 | 3 | 0 | |

Two-sided Kruskal-Wallis chi-squared test: Age and Body Mass Index, Two-sided Chi-squared test: Race/Ethnicity, sexual activity description, sexual activity frequency.
[a]Kruskal-Wallis chi-squared test.
[b]Chi-Squared test.

clustering of *A. christensenii, G. piotii, G. vaginalis, G. 1,2,3,4, G. 5,6,7,8,9, P. bivia,* and an unknown *Gardnerella* (Fig. 2E). Looking at net flux values, we see analogous inter-*Gardnerella* clustering of species clustered in mutualisms (Fig. 2F).

Based on our analysis of average mutualism benefit and average competition cost as quantified via change in flux through biomass, we found that *L. iners* and *A. christensenii* both significantly benefited from pairwise mutualism compared to low competition cost (Fig. 3A). Conversely, *H. timonensis* showed high competition cost relative to its low mutualism benefit. Our analyses showed that most of the investigated strains fell within the low to medium mutualism benefit range, with a similarly low to medium competition cost, except for one *Gardnerella* strain with significant competition cost.

Using a one-tailed t-test, followed by Bonferroni multiple test corrections, we highlighted secondary bacteria that most significantly benefited primary bacteria with which they co-occurred.

The top five most mutualistic bacteria belonged to the following *Gardnerella* clades: *G. 8* (p-value: $2.18 \times 10^{-99}$, t-statistic: 42.6), unknown strain (p-value: $4.59 \times 10^{-93}$, t-statistic: 39.2), *G. piotii B* (p-value: $7.12 \times 10^{-85}$, t-statistic: 35.2), *G. leopoldii* (p-value: $3.50 \times 10^{-84}$, t-statistic: 34.8), and *G. 5* (p-value: $4.31 \times 10^{-80}$, t-statistic: 32.9). Conversely, the top bacterial competitors were defined as secondary bacteria that most significantly outcompeted primary bacteria, resulting in a decrease in biomass during pairwise simulations. The top five most competitive bacteria fell under the following *Gardnerella* clades: *G. 8* (p-value: $4.82 \times 10^{-51}$, t-statistic: 21.3), unknown species (p-value: $3.77 \times 10^{-44}$, 18.9), *G. 9* (p-value: $1.57 \times 10^{-43}$, t-statistic: 18.7). *G. 3* (p-value: $1.74 \times 10^{-43}$, t-statistic: 18.7), and *G. 7* (p-value: $1.74 \times 10^{-43}$, t-statistic: 18.7). We overlaid the most competitive and most mutualistic bacteria onto the *Gardnerella* ANI dendrogram and observed a greater genetic diversity among top mutualistic bacteria compared to the top competitive bacterial strains (Fig. 3B).

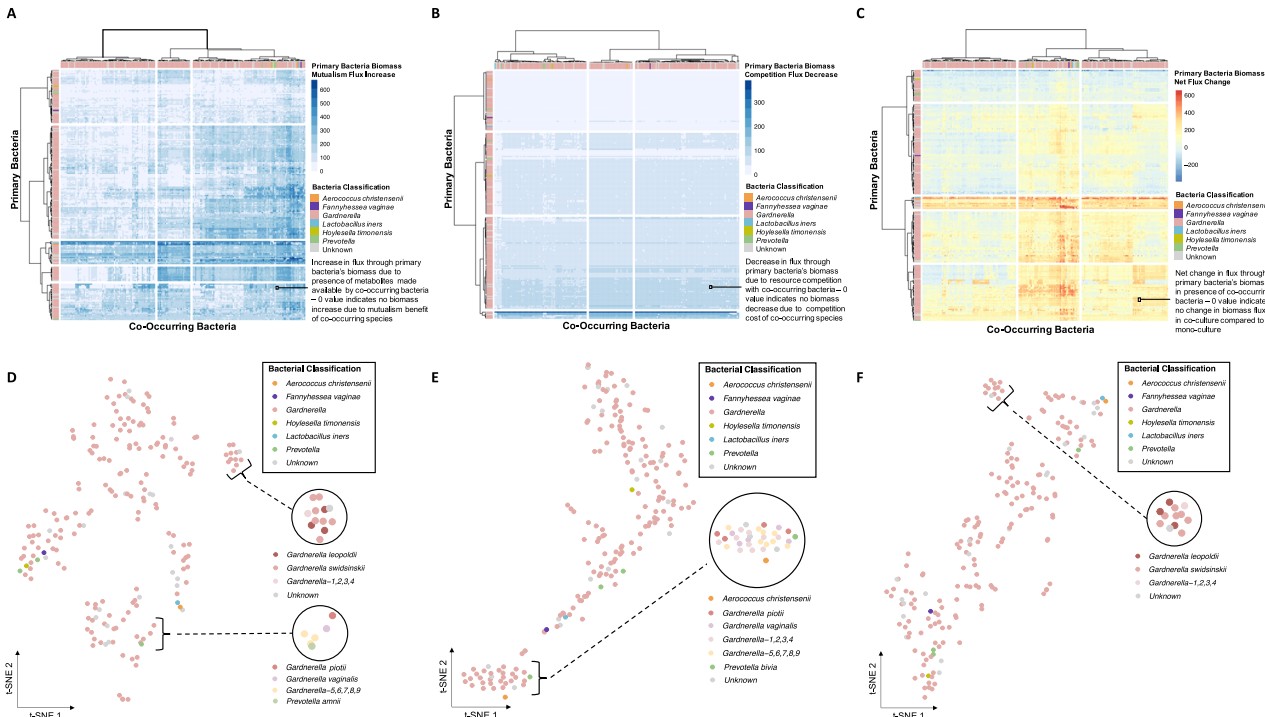

**Fig. 2 | Pairwise in silico bacterial interactions.** Heatmap illustrating primary bacteria biomass flux change in the presence of co-occurring bacteria when simulating (**A**) mutualism, (**B**) competition, (**C**) net interactions; t-SNE plots of primary bacteria biomass flux change reduced across all co-occurring bacteria when simulating (**D**) mutualism, (**E**) competition, (**F**) net interactions.

Our analysis of the most commonly shared and/or competed for metabolites highlighted the physiological relevance of four metabolites in the BV-positive in silico bacterial interaction simulation (Fig. 3C). Specifically, L-histidine (Competitive interactions: 98%; Mutualistic interactions: 94%), which can be decarboxylated to form histamine, a key host inflammatory response immune regulator (Branco et al., 2018). Selenocysteine (Competitive interactions: 94%) is an essential component of selenoproteins, which play a role in host immune function (Avery & Hoffmann, 2018; Rayman, 2000). Sphinganine 1-phosphate (S1P) (Competitive interactions: 35%) has been associated with pro-inflammatory properties and implicated in tissue damage, but rarely implicated in bacterial metabolism (Ledgerwood et al., 2008; Liu et al., 2021). Iron (III) (Competitive interactions: 78%; Mutualistic interactions: 75%) is an essential *Gardnerella* growth micronutrient (Jarosik et al., 1998). We directly probed the relationship between L-histidine and *G. piotii* growth. *G. piotii* had a 41.3% growth benefit when grown in NYCIII enriched with 0.1% L-histidine compared to NYCIII enriched media alone, as confirmed by a one-tailed t-test ($t = 3.49$, $p$-value = 0.002) (Fig. 3D).

### Experimental validation highlights metabolic interactions that drive BV-associated microbial dynamics

**Growth Curves.** Because of the fastidious nature of *P. bivia*, it was not used for in vitro analysis. *P. amnii*, *F. vaginae*, and *H. timonensis* all showed similar growth capacity in NYCIII enriched media compared to *G. piotii* spent media (Fig. 4A–C). *L. iners* and *G. vaginalis* had 44% and 37% reduced growth in *G. piotii* spent media compared to enriched media (Fig. 4D, E). All species had minimal growth in *G. vaginalis* spent media. *G. piotii*, *A. christensenii*, and *P. buccalis* had minimal growth across all three conditions (Fig. 4F–H).

**Metabolomics.** Both *P. amnii* and *G. piotii* showed no growth in *G. vaginalis* spent media and were not submitted for metabolomics

analysis. After filtering for p-value and log$_2$(fold change) (<0.01, > |2|) 85 metabolites were identified as significantly consumed and 87 metabolites were identified as significantly produced (Table 2). Using ORA we identified the top three metabolite super-classes that were most significantly enriched in both consumed and produced metabolites based on enrichment ratio. Consumed metabolites were enriched for nucleic acids (Enrichment-ratio: 272.7; Holm $p$-value: $1.95 \times 10^{-12}$), carbohydrates (Enrichment-ratio: 182.9; Holm $p$-value: $1.57 \times 10^{-5}$), and organic acids (Enrichment-ratio: 54.5; Holm $p$-value: $3.05 \times 10^{-8}$). Produced metabolites were enriched for nucleic acids (Enrichment-ratio: 263.2; Holm $p$-value: $2.0 \times 10^{-14}$), carbohydrates (Enrichment-ratio: 100.5; Holm $p$-value: 0.004), and organoheterocyclic compounds (Enrichment-ratio: 58.4; Holm $p$-value: $1.74 \times 10^{-5}$).

*G. piotii* significantly consumed L-histidine when grown in fresh media (Fig. 5A). Both *G. vaginalis* and *G. piotii* produced protocatechuic acid (PCA) and 2-hydroxyisobutyric acid (2-HIBA) (Fig. 5A, B). PCA has been shown to have beneficial anti-inflammatory effects in the gut (Murota et al., 2018). Both *G. vaginalis* and *H. timonensis* significantly consumed the volatile fatty acid, propionic acid, which has been found to be elevated in BV-positive cervicovaginal fluid (Fig. 5B, C)[3,25–27]. We directly investigated the relationship between propionic acid and *H. timonensis* growth. *H. timonensis* had a minimal growth benefit when grown in NYCIII enriched with 0.01% propionic acid compared to NYCIII enriched media alone (S2A). *F. vaginae* grown in *G. vaginalis* spent media did not have any significantly different metabolites. *L. iners* significantly produced lactic acid only when grown in *G. piotii* spent media (Fig. 5D). Caffeate is specifically produced by *F. vaginae* when grown in *G. piotii* compared to NYCIII enriched media ($p$-value = 0.07) (Fig. 5E, F). Caffeate is able to bind to estrogen receptor alpha (ERα), which is located throughout the vaginal tissue[28]. Other differentially produced or consumed metabolites were found, but their physiological relevance was not explored further (S2 B-I).

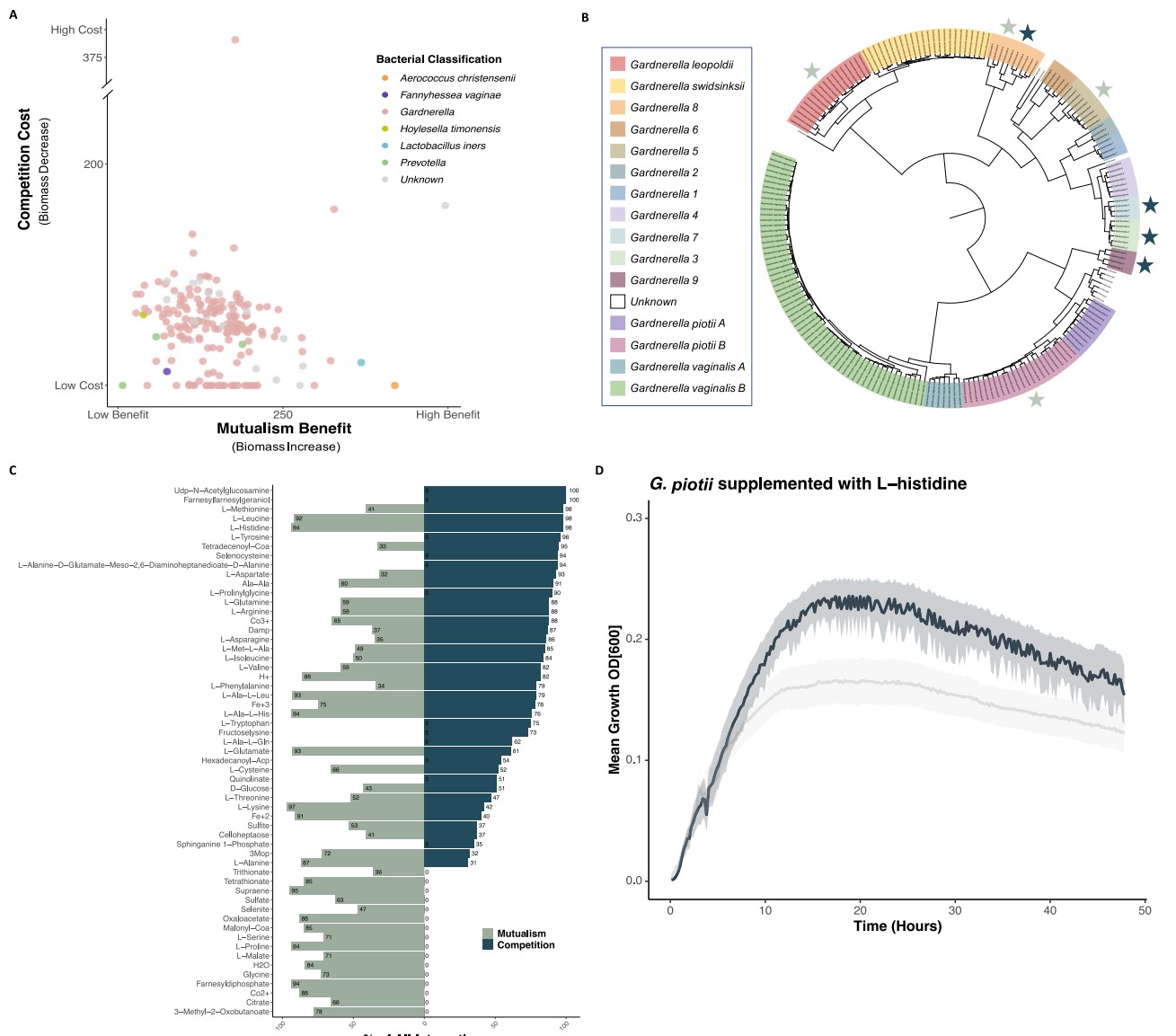

**Fig. 3 | Relative competition versus mutualism and the metabolites that underly these interactions. A** average biomass increase due to mutualism benefit vs. average biomass decrease due to competition cost for each bacteria (**B**) dendrogram of *Gardnerella* species, with light stars denoting top four most mutualistic bacteria and dark stars indicating top four most competitive bacteria across all interactions simulations. **C** Most highly competed for (competition) and/or shared (mutualism) metabolites across all interaction simulations. **D** *G. piotii* growth curves in NYC III enriched media (gray) and NYC III enriched media supplemented with 0.1% L-histidine (black), with mean and ½ standard deviation spread (one-tailed t-test, $t = 3.487$, $p$-value < 0.01).

## Discussion

Despite the significant impact of BV on both acute and chronic health, there is a lack of understanding regarding the microbial community structures and metabolic interactions underlying this condition. Our study employs clinical data, in silico, and in vitro analyses to investigate the variations in *Gardnerella* co-occurring species between symptomatic and asymptomatic BV, as well as explore the differential interactions of *Gardnerella* strains with each other and non-*Gardnerella* co-occurring species. By using spent media, we evaluate how bacterial growth is influenced by the metabolic byproducts and nutrient depletion caused by other bacteria, providing a more realistic insight into inter-bacterial dynamics. Additionally, we compare in silico predictions with in vitro metabolomics of spent media to further understand the metabolites defining inter-bacterial interactions.

Using in vivo metagenomic analysis of vaginal swabs, we observed variations in BV-associated community structure between symptomatic and asymptomatic samples. Our findings indicate a higher prevalence of *A. christensenii* and *P. bivia* in symptomatic samples, while *P. amnii, P. buccalis,* and *H. timonensis* were more frequently found in asymptomatic samples. These differences in community structures underscore the need for further sub-categorization of BV. By establishing more specific definitions, development of more targeted treatments that move away from the current broad-spectrum antibiotic approach are possible[29–31].

The complexity of BV is further supported by in silico simulations of pair-wise bacterial interactions. GENREs are a powerful tool for predicting metabolic interactions because they allow for the simulation of complex biological systems and the identification of potential metabolic pathways and interactions. By providing a framework to analyze thousands of pair-wise interactions, GENREs help uncover the metabolic dynamics that are not immediately apparent through experimental methods alone. Our analysis reveals significant

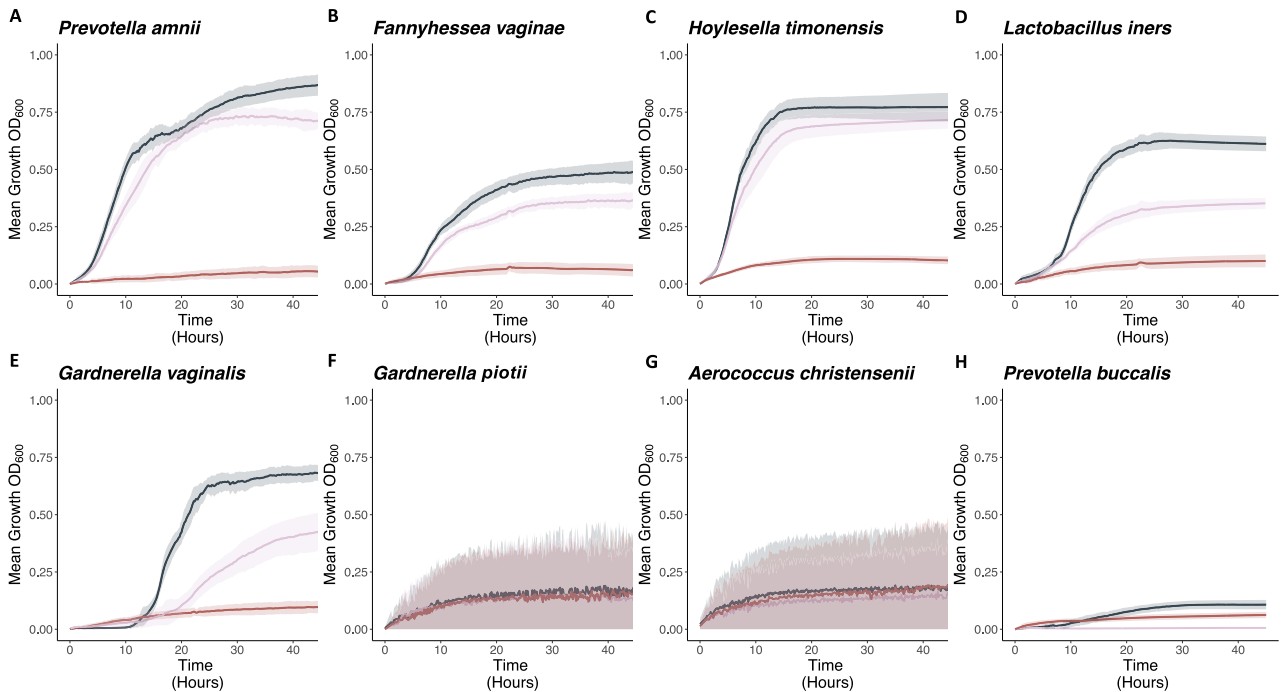

**Fig. 4 | Pairwise in vitro bacterial growth interactions. A–H** growth curves in NYC III enriched media (black), spend *G. piotii* media (light pink), and spent *G. vaginalis* media (dark pink/red), with mean and two standard deviation spread. *G. piotii* and *A. christensenii* have highly flocculent phenotypes.

clustering between *Gardnerella* species based on assessed mutualistic benefit, indicating that genetic similarity does not necessarily correlate with functional metabolic similarities. Additionally, we observe a bifurcated metabolic benefit versus metabolic cost of *L. iners*, all *Prevotella* species, *H. timonensis*, *F. vaginae*, *A. christensenii*, and certain *Gardnerella* strains when co-occurring with *Gardnerella* strains. This result suggests that a strain's metabolic relationship may be beneficial in one inter-bacterial context but detrimental in another. Overall, *A. christensenii* and *L. iners* appear to consistently benefit the most from inter-bacterial interactions in silico. Experiments reveal that *A. christensenii* does not experience growth benefits when cultured in spent media of either *G. vaginalis* or *G. piotii*. These findings differ from our in silico predictions and demonstrate that pair-wise simulations cannot

fully capture the complexity of the polymicrobial communities present in BV.

We investigated what specific metabolites were being consumed and produced by co-occurring BV-associated species. *G. piotii* significantly consumed L-histidine both in our metabolomics analysis and growth curve analysis, which recapitulated our in silico prediction that L-histidine is highly competed over. Our metabolomics analysis was also able to reveal metabolite production and consumption that were not demonstrated in our in silico analysis. Specifically, *F. vaginae* significantly produced caffeate when grown in *G. piotii* spent media, which we confirmed is specific to a BV-associated metabolic context and is not ubiquitously produced. Previous studies have shown that caffeate can bind to ER$\alpha$, which regulates estrogen sensitive gene expression. Reduced circulating estrogen levels results in vaginal epithelial atrophy, as seen in post-menopausal women[32]. These findings point to the importance of ER$\alpha$ in supporting vaginal epithelial health. Conversely, previous research has demonstrated estrogen-dependent focal adhesion kinase (FAK) activation in vaginal epithelial cell tissue. The mechanism of action centers on estrogen binding to ER$\alpha$, precipitating ER$\alpha$ phosphorylation, which cascades to FAK phosphorylation. Additionally, it was found that estrogen dependent FAK activation precipitates increased bacterial adhesion on the vaginal epithelium[33]. Lastly, previous research has shown BV-associated metabolites can alter vaginal immune response[34,35]. These data in combination with our data, indicate that BV-associated metabolic byproducts may alter vaginal epithelial regulation.

In summary, our study sheds light on the varied community composition and metabolic interactions associated with BV, a prevalent vaginal condition that significantly impacts women's health. By analyzing vaginal metagenomic data and constructing pair-wise metabolic network simulations of BV-associated bacteria, we highlighted the context dependent nature of these competitive and mutualistic relationships. Through metabolomics analysis, we further investigated the role of specific metabolites in these inter-bacterial interactions. Our findings further emphasize the complex and diverse nature of BV-associated bacterial community structures. While our study provides valuable insights using relative abundance via

**Table 2 | Metabolites identified as significantly consumed or produced based on spent media metabolomics (p-value < 0.01(two-sided t-test, Bonferroni multiple hypothesis correction); log2(fold change)>|2|)**

| Metabolomics Comparison | Consumed | Produced | Total |
|---|---|---|---|
| *A. Christensenii* in *G. piotii* Spent Media | 4 | 11 | 15 |
| *P. buccalis* in *G. piotii* Spent Media | 9 | 3 | 12 |
| *F. vaginae* in *G. piotii* Spent Media | 5 | 12 | 17 |
| *H. timonensis* in *G. piotii* Spent Media | 15 | 12 | 27 |
| *L. iners* in *G. piotii* Spent Media | 9 | 6 | 15 |
| *G. piotii* in Fresh Media | 6 | 7 | 13 |
| *P. amnii* in *G. piotii* Spent Media | 10 | 11 | 21 |
| *G. vaginalis* in *G. piotii* Spent Media | 11 | 9 | 20 |
| *A. christensenii* in *G. vaginalis* Spent Media | 1 | 0 | 1 |
| *P. buccalis* in *G. vaginalis* Spent Media | 3 | 4 | 7 |
| *H. timonensis* in *G. vaginalis* Spent Media | 3 | 0 | 3 |
| *L. iners* in *G. vaginalis* Spent Media | 3 | 0 | 3 |
| *G. vaginalis* in Fresh Media | 6 | 12 | 18 |
| Total | 85 | 87 | 172 |

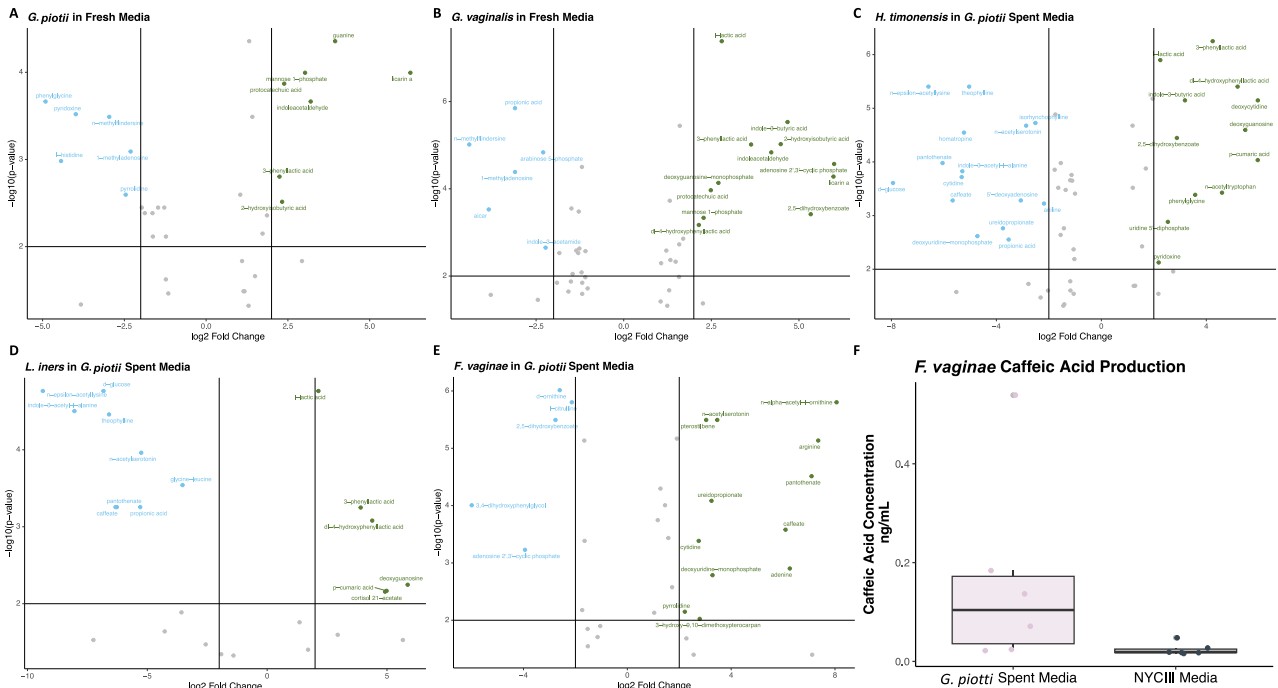

**Fig. 5 | Pairwise in vitro bacteria metabolic interactions. A–E** Volcano plots of differential metabolites (blue: consumed; green: produced; gray: not significant (two sided t-test, Bonferroni multiple hypothesis correction) from co-occurring species grown in *G. vaginalis* spent media (**F**) Caffeic acid production of *F. vaginae* grown in *G. piotii* spent media versus NYC III ($n$ = 6 biological replicates, $p$-value = 0.07 (one-sided t-test) *G. piotii* Spent Media boxplot: minimum = 0.022, maximum = 0.539, median = 0.104, 25th percentile = 0.024, 75th percentile = 0.184, upper whisker = 0.184, lower whisker = 0.022, interquartile range = 0.16. NYC III Media boxplot: minimum = 0.016, maximum = 0.048, median = 0.0185, 25th percentile = 0.018, 75th percentile = 0.027, upper whisker = 0.027, lower whisker = 0.027, interquartile range = 0.009.

metagenomics, it is important to note that this approach may not accurately reflect the absolute bacterial loads, which could be more precisely quantified using qPCR. Moreover, while our in vitro growth experiments provide valuable insights into bacterial metabolic interactions, they may not fully capture the complexity of in vivo dynamics, particularly host-microbiome interactions involving immune responses and epithelial cell function. This consideration underscores the need to develop intricate simulations that go beyond pairwise interactions to more accurately reflect in vivo community dynamics. Further research in this field will lead to improved treatments for BV, as well as a deeper understanding of women's health.

## Methods
Our research complies with all relevant ethical regulations and was conducted in accordance with protocols approved by a federally accredited Institutional Review Board (Viome IRB# 20220118.evvy).

### *Gardnerella* and co-occurring species comparison
**Sample collection and cohort characteristics.** For this present study, we analyzed metagenome sequencing data collected from participants using the Evvy platform in the United States. All participants provided informed consent for the use of samples and data for this present study, and study procedures were conducted in accordance with protocols approved by a federally accredited Institutional Review Board (Viome IRB# 20220118.evvy). The Evvy platform workflow begins when a patient requests a test from their healthcare provider, who will then submit a lab order for an Evvy vaginal health test. A sample collection kit is then shipped directly to the patient. The patient self-collects a vaginal swab using Copan eNAT collection tubes (Copan, Murrieta, CA, USA), and ships the sample at ambient room temperature to Microgen DX laboratory (Lubbock, TX, USA) which is subsequently processed using a validated CLIA/CAP/CLEP certified

shotgun metagenomics pipeline within five days. Following metagenomic sequencing, the Evvy workflow then employs a specialized bioinformatics pipeline to analyze the metagenome data. The result is a detailed report on the presence and relative abundances of 700+ microbes which can then be reviewed by the patient's healthcare provider to develop a care plan. More details on the metagenome data processing and analysis are available in the following section.

Each participant also completed a questionnaire capturing symptoms, diagnoses, and demographic information. Participants in the study cohort included individuals of varying ages, ethnic groups, sexual orientations, and sexual activity levels, all of which gave informed consent to participate in the study. Since all samples collected were vaginal in nature, the sex of all participants was female. Table 1 describes the demographics and clinical characteristics of the study population in more detail.

**Metagenome data processing and analysis.** For the shotgun metagenomic sequencing workflow, vaginal samples were processed using a host depletion step with PMA, followed by chemical lysis with metapolyzyme (Sigma, St. Louis, MA, USA), bead beating, and DNA extraction via KingFisher FLEX automated extraction (Thermo Scientific, Waltham, MA, USA). Negative and positive controls (NC: water, PC: ATCC MSA-2007 mock community) were included with each batch of extractions to ensure data integrity. Sequencing libraries were prepared from the extracted DNA and sequenced on an Illumina NovaSeq platform. Raw sequencing reads were processed using the Evvy bioinformatics pipeline, which includes quality control via Trimmomatic to remove low-quality bases and host DNA depletion through mapping to the GRCh38 human genome reference. All host-depleted raw sequencing reads (fastq files) have been deposited into the sequence read archive (SRA) can be found at the following BioProject accession number: PRJNA1219227. Microbial taxonomic classification

was performed using a database of more than 4000 microbial genomes, predominantly sourced from the urogenital tract, with supplemental genomes from other human microbiome sites to ensure phylogenetic diversity. Taxa with a relative abundance of >0.75% were included in further analyses. The database used for classification was constructed from publicly available genome repositories (NCBI and EBI) and previously published metagenome-assembled genomes generated from Evvy's test data (Supplementary Data 1). No de novo MAGs were assembled as a result of this present study.

**Co-occurrence analysis.** For this present study, we analyzed the metagenome sequencing data previously collected[36] which has been made publicly available (See data availability statement). Samples for co-occurrence analysis were selected based on self-reported symptom profiles. In the absence of microscopy based diagnostics, Amsel Criteria and Nugent score, BV-associated profiles were defined by *Gardnerella* dominance via metagenomic sequencing. *Gardnerella*-dominant profiles were selected and then stratified by either asymptomatic or symptomatic, defined as the presence of self-reported excessive discharge and vulvovaginal pruritus.

Samples were split into three different groups: symptomatic BV ($N = 91$), asymptomatic BV ($N = 489$), and a healthy cohort ($N = 122$). The symptomatic and asymptomatic BV groups both consisted of samples of Community State Type (CST) 4 and were dominated by *Gardnerella* (>= 50% relative abundance)[37]. However, the symptomatic BV group experienced excessive discharge and itchiness (either internal or external), while these symptoms were absent in the asymptomatic BV group. The healthy cohort consisted of samples of CST 1, 2, or 5, dominated by lactobacilli, with a relative abundance of ≥50% *L. crispatus, L. gasseri, L. paragasseri, L. pensenii, L. mulieris,* and did not experience excessive discharge nor itchiness. Relative frequency of species that co-occur with *Gardnerella vaginalis* within each subgroup was calculated using the following formula:

# of samples with ≥2% relative abundance of co-occurring species
# of samples with ≥2 % relative abundance of *G. vaginalis*

**Average nucleotide identity for speciation determination.** The nucleotide similarity between the *Gardnerella* genomes was determined using ANI comparison[38]. 222 *Gardnerella* genomes were input to pyani (v0.2.12), a Python implementation tool for ANI analysis (Pritchard et al., 2015). After running an initial comparison with pyani using the ANIb method, 13 genomes were removed from the analysis as they were too divergent from the majority of genomes. The remaining 209 *Gardnerella* genomes were input to pyani, and the ANIb method which aligns 1020 nt fragments of the input sequences using BLASTN+ was used[39,40]. The generated percentage identity heatmap was color coded based on the taxonomic classifications given by the NCBI records of the genomes[41].

**Dendrogram construction.** The pyani analysis generated a distance matrix for percentage identity, which was used to generate a dendrogram. The distance matrix was clustered via hierarchical clustering using the "complete" method via the TAPE R package (v0.3.0)[42]. The clustered output was then converted to a tree object in Newick tree format. The dendrogram tree file was visualized using iTOL v6.7.1 and color coded based on the taxonomic classifications of the genomes' NCBI records[43].

**Model construction and contextualization.** We analyzed potential metabolic interactions among *Gardnerella* strains by identifying 208 *Gardnerella* whole genome sequences from the BV-BRC 3.28.21 database that meet the "good" quality criteria defined as a genome with at least 80% completeness, less than 10% contamination, and at least 87% consistency with known protein sequences[44]. Thirteen genomes were removed due to ANI clustering indicating either incomplete genome

sequence or chimera status. We used the remaining 195 *Gardnerella* strains' annotated amino acid sequences to generate GENREs using the Reconstructor (v1.1.0) algorithm[45]. Additionally, we created GENREs for seven co-occurring species of interest: *Lactobacillus iners, Prevotella bivia, Prevotella amnii, Prevotella buccalis, Hoylesella timonensis, Fannyhessea vaginae,* and *Aerococcus christensenii.* The quality of the GENREs was assessed using MEMOTE (v0.13.0), which is the standard metric in computational metabolic modeling[46]. The GENREs and corresponding MEMOTE scores are available at the project's GitHub repository[47].

We identified the top five most competitive and most mutualistic bacteria using a one-tailed t-test and corresponding Bonferroni multiple test correction using the p.adjust R function from Rstats package (v3.6.2)[48]. Finally, we calculated the percent of total interactions a metabolite was either competed for and/or shared between GENREs.

## Spent media analysis

**Metabolomics sample collection.** The following describes the experimental setup and sample preparation for metabolomics analysis of *Gardnerella vaginalis* (strain ATCC 14018) and *Gardnerella piotii* (strain JCP8151B) primary spent media. The bacteria were grown overnight under anaerobic conditions in NYCIII + 2% FBS media, and inoculated into T25 polystyrene tissue culture flasks. After 18 h of incubation, samples were collected from each flask, spun down (15 min; 6918 g), and filter-sterilized (0.2 μ filter). The filtered supernatant was stored at −80 °C until metabolomics processing. $OD_{600}$ was recorded prior to the spin down step. Pooled spent media samples for both *G. vaginalis* and *G. piotii* were also aliquoted into microcentrifuge tubes and stored at −80 °C. The remaining spent media for *G. vaginalis* and *G. piotii* was stored at −20 °C.

To collect metabolomics on spent media conditions, six of the co-occurring species—*L. iners, P. amnii, P. buccalis, H. timonensis, F. vaginae* and *A. christensenii*—were grown overnight in enriched media (NYCIII + 2% FBS, BHI, or MRS) under anaerobic conditions. Twelve-well plates with 2.5 mL of either *G. piotii* or *G. vaginalis* spent media were inoculated with 100 μL of co-occurring species inoculum. After inoculation, samples were grown to saturation and $OD_{600}$ measured. Samples were then collected from each flask, spun down (5 min; 3075 g), filter-sterilized and stored at −80 °C.

**Mass spectrometry sample preparation and analysis.** Untargeted metabolomics was performed by the University of Virginia Biomolecular Analysis Facility Core. Sample preparation involved adding 2000 μL of 80% methanol (−20 °C) to 500 μL of each culture medium. The mixture was vortexed for one minute and incubated at −20 °C for 2 h to induce protein precipitation. Subsequently, the tubes were centrifuged at $14,000 \times g$ for 30 min at 4 °C. The resulting supernatants were transferred to new tubes, and a quality control (QC) sample was prepared by combining 35 μL from each sample. All samples were then dried in a speed vacuum for approximately four hours and stored at −80 °C until further processing. There were $N = 5$ biological replicates for each mass spec sample. In total, we analyzed 79 samples, including 12 experimental conditions ($n = 5$ per condition), and 5 control conditions (*G. vaginalis* primary spent media ($n = 5$), *G. piotii* spent media ($n = 5$), *G. vaginalis* pooled primary spent media ($n = 2$), *G. piotii* pooled primary spent media ($n = 2$), and NYCIII blank media ($n = 5$)).

For mass spectrometry analysis, the dried samples were reconstituted in 100 μL of 0.1% formic acid along with 100X diluted Metabolomics QReSS heavy-labeled standards (Percy et al., 2022; Percy, Andrew et al., 2021). A dilution of 1:800 was applied to both samples and QC sample. Subsequently, 10 μL of each diluted sample was injected for analysis.

Mass spectrometry data acquisition was performed using a Thermo Orbitrap IDX MS connected to a Vanquish UPLC system. A Waters BEH C18 column ($100 \times 2.1$ mm, 1.9 μm) was employed for soluble metabolite separation. The column temperature was set at 30 °C, and a flow rate of 250 μL/min was maintained. The mobile phase consisted of 0.1% formic acid in water (mobile phase A) and 0.1% formic acid in methanol (mobile phase B). The mass scan range was set from 67 to 1000 with a resolution of 120,000 and a scan range of 0.6 s.

Data-dependent MS2 scans were obtained using a real-time precursor exclusion strategy with AcquireX mode. Prior to the analysis of the samples, initial runs were performed with solvent blank samples in full scan mode (67–1000 mass range) to generate an exclusion list containing all the peaks present in the blank samples. A QC sample was then run in full scan mode, and all the peaks were saved as an inclusion list. Four data-dependent acquisition (DDA) injections were performed, with the exclusion list automatically updated based on the selected precursor.

After data acquisition, the samples were subjected to analysis using the open-source software MS-DIAL[49,50]. To identify and remove background ions, five blank samples were included in the analysis. The samples acquired in full scan MS1 mode were used for quantification, while the samples acquired in DDA mode were utilized for spectral identification of metabolites. The MS1 tolerance was set to 0.01 Da, and the MS2 tolerance was set to 0.025 Da. For peak picking, a mass slice width of 0.1 was employed. Peak alignment was performed with a maximum retention time tolerance of 0.2 min and an MS1 tolerance of 0.015.

Peak identification was carried out by searching the MS2 spectra against the MS-DIAL public database (January 2023). A mass tolerance of 0.01 Da for MS1 and 0.05 Da for MSMS was applied, with an identification score cutoff of 60%. Additionally, the peaks were searched against an in-house IORA library in both positive and negative mode, using a mass tolerance of 0.01 Da and an identification cutoff of 80%. The data was manually inspected, and identifications without corresponding MS2 spectra were filtered out, except for IORA (ST1 & ST2). Data analysis was performed using Metaboanalyst 5.0[51]. The samples were normalized by median and subjected to log transformation. Subsequently, a fold change analysis was performed to detect dysregulated metabolites between secondary spent media compared to primary spent media. Statistical analysis was conducted using a t-test to determine the significance of group differences. Over representation analysis (ORA) was conducted via Metaboanalyst, which calculates enrichment ratio derived from comparing between actual versus predicted number of metabolite hits per super-class metabolite category[51]. The statistical significance of the enrichment ratios is calculated using hypothesis testing using a binomial distribution to assign a $p$-value, which is adjusted using Holm's multiple comparison test[52]. Data was visualized using volcano plots generated in R's ggplot2 package, with a p-value cutoff of <0.01 and a $\log_2$(fold change) cut off of >|2 | .

**Targeted caffeic acid quantification.** *Fannyhessea vaginae* culture was grown in NYCIII + 2%FBS for 24–48 h at 37 °C under anaerobic conditions. Active cultures were then diluted and transferred to fresh NYCIII + 2% FBS or filter-sterilized spent *Gardnerella piotii* media for 24 h. Cultures were then centrifuged to pellet cells, and supernatants filter sterilized (0.22 μ filter) and submitted for measurement of caffeic acid using mass spectrometry by the Biomolecular Analysis Facility at UVA.

Initially, 250 μL of each culture media was treated with 1000 μL of 80% methanol kept at −20 °C, followed by vortexing for 1 min and incubation at −20 °C for 2 h to induce protein precipitation. Subsequently, the samples underwent centrifugation at 4 °C and $14,000 \times g$ for 30 min, after which the resulting supernatants were transferred to new tubes. These samples were then dried in a speed vac for

approximately 5 h and stored at −80 °C until further analysis. Before mass spectrometry analysis, the dried samples were reconstituted in 50 μL of 0.1% formic acid, and 10 μL of each reconstituted sample was injected for analysis.

For generating the standard curve, a stock solution of the target molecule, caffeic acid, was prepared at 1 mg/mL in 80% methanol and subsequently diluted down to 1000 ng/mL in 0.1% formic acid. Serial dilutions were carried out to generate standard curves ranging from 100 ng/mL to 0.01 ng/mL. For analysis, 10 μL of each standard was injected.

The mass spectrometer settings were configured with a full scan parameter, including MS1 resolution of 120,000, scan range from 67 to 1000 m/z, RF Lens set at 60, AGC at 25%, and a maximum injection time of 50 ms, all conducted under negative polarity. Additionally, Parallel Reaction Monitoring (PRM) parameters were employed, featuring a collision energy (CE) of 30, resolution set at 50,000, an isolation window of 1.5 m/z, AGC at 50%, and a maximum injection time of 86 ms, also under negative polarity. The mobile phases used were Phase A: 0.1% formic acid in water and Phase B: 0.1% formic acid in 90% methanol.

The obtained spectra were processed and analyzed using the Skyline software. In this analysis, transitions specific to caffeic acid were included, and the retention time and peak picking were manually assessed for accurate peak area detection. The calculated concentrations were derived from the linear regression equation of the standard curve. To obtain the final concentration of the samples, the calculated values were divided by the sample concentration (5x). This comprehensive methodological approach ensures the accurate and reliable quantification of caffeic acid in the given samples.

**Growth analysis.** Overnight cultures of *L. iners, P. amnii, P. buccalis, H. timonensis, F. vaginae, A. christensenii, G. vaginalis,* and *G. piotii* were cultured under anaerobic conditions for 24–48 h in NYCIII media + 2% FBS, $OD_{600}$ measured, cells pelleted by centrifugation (2796 g, 5 min) and resuspended in fresh NYCIII + 2% FBS. Cultures were then diluted in spent media and aliquoted into a 96-well plate (starting $OD_{600}$ ~ 0.05) containing spent or blank media for a kinetic growth assay under anaerobic conditions without shaking. *G. piotii* and *A. christensenii's* flocculant phenotype necessitated the use of shaking. There were $N = 7$ replicates per condition for the growth curve data which were distinct samples.

**Compound-specific growth analysis.** To directly probe the relationship between L-histidine and *G. piotii* growth, *G. piotii* was cultured under anaerobic conditions for 48 h in NYCIII + 2% FBS media. $OD_{600}$ was measured and bacteria were pelleted via centrifugation (2795 g, 5 min) and resuspended in NYCIII media. Cultures were then diluted in NYCIII media and aliquoted into a 96-well plate (starting $OD_{600}$ ~ 0.05) containing either NYCIII or NYCIII + 0.1% filter sterilized L-histidine for a kinetic growth assay under anaerobic conditions with shaking. We quantified growth differences via one-tailed t-test at the NYCIII condition's max growth time point, in addition to calculating percent growth difference between conditions.

To assess the influence of propionic acid on the growth kinetics of *H. timonensis*, a kinetic growth assay was carried out under anaerobic conditions. Propionic acid stocks at a 2x concentration were freshly prepared, filter-sterilized, and anaerobically reduced. Each well of the plates received 100 μL of *H. timonensis* culture and 100 μL of NYCIII/ propionic acid solution, resulting in final propionic acid concentrations of 1%, 0.1%, and 0.01% (w/v). Anaerobic conditions were maintained using a breathe-easy membrane, and the plate was agitated at 31 g with a shaker that held a Cerillo stratus. Periodic OD600 readings were taken using the Cerillo Stratus to monitor growth kinetics, all normalized to the initial OD reading.

### Statistics and reproducibility

Statistical analyses were performed in several instances in this study to determine significant differences between groups of interest. We provide information about statistical tests and sample size when reporting a p-value to ensure transparency in how data was analyzed and to allow for reproducibility of the study. No data points were excluded in any of our statistical analyses. No statistical method was used to predetermine sample size. The investigators were not blinded to allocation during experiments and outcome assessment. We further ensure reproducibility of our computational experiments and analyses by providing all necessary code and input files. Additionally, we provide a detailed description of the functionality of all scripts and inputs and outputs from each script in the readme file in the GitHub repository (https://github.com/lrd3uu/bacterialvaginosis_interactions).

### Reporting summary

Further information on research design is available in the Nature Portfolio Reporting Summary linked to this article.

## Data availability

The raw metabolomics files provided to us by the UVA Metabolomics core were uploaded to the Metabolights repository under the following identifier: REQ20250415209971. Stable dropbox links to all raw metabolomics files are also available in the GitHub repository in the readme file under the metabolomics heading (https://github.com/lrd3uu/bacterialvaginosis_interactions). The host-depleted raw metagenomic sequencing data (fastq files) are available on the Sequence Read Archive (SRA) under the following BioProject accession number: PRJNA1219227). All MAGS used in this study were previously made publicly available and can be accessed via BioProject accession numbers listed in Supplementary Data 1.

## Code availability

Code, input files, and output files are available in a public Github repository: https://github.com/lrd3uu/bacterialvaginosis_interactions. All metabolic network models generated during this study are available in a public Github repository: https://github.com/emmamglass/Gardnerella-Interactions. For citability and reproducibility purposes, we have created a stable doi identifier for this repository via zenodo: https://doi.org/10.5281/zenodo.15231789.

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

## Acknowledgements

We thank the Biomolecular Analysis Facility in the University of Virginia School of Medicine. for mass spectrometry services. This work was supported by the National Science Foundation. (GRFP award number 1842490 to EG, NRT-ROL 2021791 to LD), the National Institutes of. Health (1 T 32 GM 145443-1 to EG, R01-AI154242 to JP, R01-AT010253 to JP) and the. National Institute of General Medical Sciences (5T32GM136615-03 to LD), and Evvy (to LD).

## Author contributions

L.D. wrote initial manuscript draft. L.D., E.G., R. M., D. L. and F.W. performed computational studies, L.D. and G.K. performed experimental analyses, K.T.-W. and J.P. supervised the work, L.D., G.K., K.T.-W., F.W., R. M., D. L., E.G. and J.P. edited and approved of the final manuscript.

## Competing interests

Papin has financial stake in Cerillo, the manufacturer of the plate reader used in some experimental analyses. The initial clinical cohort was funded and conducted by Evvy. Thomas-White, Wever, Markowitz, and Lyttle are employed by Evvy, and Dillard received partial stipend support from Evvy. The remaining authors declare no competing interests.
