## [Transparent Peer Review file · Nature Communications]

Genome-scale metabolic network reconstruction analysis identifies bacterial vaginosis-associated metabolic interactions

Corresponding Author: Professor Jason Papin

Version 0:

Reviewer comments:

Reviewer #1

(Remarks to the Author)

Overall, this manuscript describes an interesting attempt to look at the interactions among different bacterial strains that are commonly found in the healthy and less healthy vaginal biome. Aside for the absence of key components of the manuscript (e.g., figure legends), there are some very interesting outcomes. Thus, the biomass flux analyses are unique and do lead to the conclusions described. Differential clustering of the multiple *Gardnerella* genome species is interesting although the mechanisms are unclear. The co-culturing experiments (e.g., in spent media from different taxa) are interesting in that they may begin to get at the mechanisms of the exclusion and mutualism observed. The mass spec identification of ~85 metabolites was also interesting, and the focus on a couple of the compounds that were found in more than one taxon followed up by testing their effects in vitro was a major plus. A concern of the manuscript is how the taxa were selected. Probably the most important taxon omitted from the study was *Lachnocurva vaginae* (previously BVAB1). That said, the authors may be excused of this since BVAB1 is difficult/impossible to culture as of yet. The absence of *Lactobacillus crispatus* in the analysis was also noteworthy. This taxon is by most measures a hallmark of a healthy vaginal ecosystem. However, since this manuscript was focused on bacterial vaginosis, it is maybe understandable that there was almost no work on *Lactobacillus crispatus*, as *L. crispatus* is almost never associated with BV. Another concern was that despite an extensive Methods section, the methodology for analyzing bacterial biomass seems to be lacking, possibly based on metagenomics, although the details are lacking as are most of the details of the microbiome characterization.

This reviewer did not have access to the information on the cohort, SOPs for collection of samples, processing of samples, etc. There is a reference to De Seta et al. 2019. It would be helpful to have a brief outline of the cohort, how they were selected, what were inclusion/exclusion criteria, how they were sampled (clinician, self-sample, ?), pregnant/nonpregnant, where in cycle, etc.

It is unfortunate that the methodology for assessing the microbiome profiles was not described. Apparently, the analysis was of metagenomic sequences, as was briefly mentioned in the Discussion section of the manuscript. However, no data was presented concerning the sample processing, library generation, sequencing technology, the sequencing strategy (Illumina short read 2 x 150 b PE??), the number of reads per sample, post sequencing data processing, analysis, etc.

Lines 77-85 and Figure 1. (Note, a significant challenge in reviewing this manuscript was the apparent lack of Figure legends. This reviewer did not have access to them). While the co-occurrence of bacterial taxa is interesting, some measure of absolute abundance, might be more relevant. The current analysis is based relative abundance, but it seems the analysis would be better if based on accurate biomass measurements. However, metagenomics is probably not an ideal strategy to measure in vivo biomass, although with some strategies it might be somewhat informative. That would depend on careful sample collection, which was not described. There is no evidence that this was considered in this study. It is possible to get reasonably accurate measures using qPCR or related technologies. However, even with qPCR, the results vary with the sample that is collected on the swab. Hence, swabs from women with BV may have more total (and more bacterial) biomass than healthy women. It is not clear to this reviewer how this might change the results.

Paragraph lines 86-90 seem to suggest that this team first identified '13 genetically distinct species'. This is not new. If they

found all 13 species in their data sets, that should be made clearer.

This reviewer finds the heat maps on Figure 2 nearly impossible to interpret. We take it on confidence that the relationships were as described in the manuscript. It's not clear if they heat maps could be presented in a better way. Minimally, it would be helpful to have a figure legend.

The growth curves in Fig.4 show some interesting interactions between spent media from bacterial cultures and growth rates of other bacteria. In most cases, the effect is negative. Interestingly, *G. vaginalis* seems to have the greatest impact.

Conclusions. This manuscript represents a significant effort to begin to identify the mechanisms by which bacterial taxa in the female reproductive tract interact positively or negatively. The results seem robust and likely to be reproducible in other cohorts. One common suggestion by reviewers is to test results in an alternate cohort. The major concern here is basing the biomass comparisons on apparently relative abundance readings. Accurate readings of absolute abundance would provide more accurate measures of biomass and could lead to some very significant differences in outcomes.

(Remarks on code availability)

I'm not a bioinformaticist. My reading of the code would not be informative.

Reviewer #2

(Remarks to the Author)

In this study, we conducted an analysis of metagenomic data obtained from human vaginal swabs in order to ascertain the microbial composition of both symptomatic and asymptomatic cases of BV. To validate these predictions, they carried out in vitro growth experiments.

The analysis is a little hard for the general reader to follow. Please clarify the following points.

Introduction

How relevant are in vitro growth experiments to in vivo dynamics? Please clarify in discussion.

Define in introduction- Average nucleotide identity (ANI)

What is GENREs... define it in intro and methods. Discuss relevance in discussion.

Need legends for all figures

Results

Clarify what is "Number of Tests" and "Number of Users" in Table 1.

How was co-occurrence calculated? (line 251)

Define pyani and AN1b analysis. (line 270)

What is t-SNE on the axis in Fig2 D, E, F. They should be defined in Methods. Explain t-SNE (line 318)

Need a clearer explanation what is being shown in Figure 2

Need a clearer explanation of what is being shown in figure 3

I assume Figure 4 are growth curves... need clarification.

Clarify the utility of using spent media?

Colors in Figure 5 are unreadable. I assume these are volcano plots?

List Compounds for table 2 in appendix.... They would be of interest to the reader

(Remarks on code availability)

Reviewer #3

(Remarks to the Author)

This paper by Dillard et al is a data analysis project of metagenomics and metabolomics datasets from vaginal swab specimens to infer mutualistic and competitive interactions between bacterial species. The paper shows various the

statistical and mathematical models used to identify relationships between vaginal bacteria and metabolites, and then tests some of these relationships using in vitro methods and co-culture. The experiments are not clearly outlined in the results from one section to the next, making it difficult to understand the source of the data (in vitro? In vivo?) with the accompanying figures. The figures do not include legends. The data on the study participants is absent in critical areas including how they define BV, which is one of the major variables in these analyses. It is also unclear what is the source of the metagenomic data? (is this published? Did they recruit participants?). Overall, this paper content is difficult to understand, misses critical information, and therefore makes it challenging to comment on the significance of this study.

Specific comments:

- There is no description of the study participant recruitment and consent information, nor the clinical and lab criteria for the definition of BV, which is a critical component of the analysis in this paper. Was this scored by Nugent, Amsel's, etc? How are they defining asymptomatic BV? In the absence of these data it makes it challenging to evaluate the downstream data analysis.
- There are no figure legends making it difficult to understand the content of the figures
- Table 1 is confusing. What is the data in the first two rows? Participants? What are 'tests'?
- What is the experiment for Figure 2 in the results section?
- While I am sure the matrix plots of figure 2 are impressive, what is the data supposed to represent? What is on the y and x axes? It looks like a comparison of Aerococcus vs Aerococcus based on the color legend.

(Remarks on code availability)

Version 1:

Reviewer comments:

Reviewer #1

(Remarks to the Author)

This is an interesting step toward understanding the interactions among vaginal bacteria and their contributions to bacterial vaginosis (BV). It underlines the likely conclusion that BV is not a single disease entity, but rather is more complex and requires more nuanced definition. Overall, although much more work is required, the data presented support the general conclusions, and the work represents a significant contribution to the field.

The response to the initial reviews is appropriate and comprehensive.

(Remarks on code availability)

Reviewer #2

(Remarks to the Author)

The authors have adequately addressed the prior review.

(Remarks on code availability)

Reviewer #3

(Remarks to the Author)

While this reviewer asked for this information in the first round, the authors still did not provide sufficient information about the study population, sampling procedures, clinical criteria, etc. In their resubmission they provided only an additional 1-2 sentence description on the clinical cohort which references a pre-print article. This is unacceptable for a journal publication. All of the information on this cohort should be provided in a standalone manner. For example, their definition of BV is flawed. They are representing the self-report of vaginal symptoms as the same as 'BV', which is not necessarily BV (it could be any of a long list of gynecological conditions). It is not clear who gave this information? The participant? The clinician?. The inclusion of a robust description of the clinical population, clinical or laboratory criteria for definition of BV, the collection and storage of biospecimens, the processing of samples, are all an absolute requirement in my opinion. Perhaps the authors should consult with the physician/gynecologist who was the clinical lead in this study to review this information for this publication to ensure its accuracy.

(Remarks on code availability)

Version 3:

Reviewer comments:

Reviewer #3

(Remarks to the Author)

Thank you to the authors for addressing my previous comments. The manuscript is significantly improved with the inclusion on information of the study cohort. This was a major gap in the previous version and having this as a component of the paper greatly improves the quality of the paper.

I have a few minor comments that I think can be quickly addressed with consultation with the editor:

- The BV definition: I appreciate the clarity in how the authors clearly how they are defining 'BV' as "In the absence of microscopy based diagnostics, Amsel Criteria and Nugent score, BV-associated profiles were defined by Gardnerella dominance via metagenomic sequencing (further details in Methods)". However, I would recommend that they use the more modern definition "molecular-BV". This term was adopted after a vigorous discussion by a consortium of scientists and OBGYN in the area of BV research to help refine these terminologies such that we have a consistent nomenclature. The reference for this discussion was in McKinnon et al, "The Evolving Facets of Bacterial Vaginosis: Implications for HIV Transmission", AIDS Res and Human Retroviruses, 2019. doi: 10.1089/AID.2018.0304. This would then remove any ambiguity about whether this is Nugent-BV, Amsel-BV, or molecular-BV.

- Symptoms in Table 1. Could the list of symptoms be listed in the Materials and Methods that are being referenced for BV? I could not find them in the manuscript. Please include them in the Results section where the cohort is first described as well as the methods section.

- information of the source data needs some improvement, and we've provided suggestions in the other section.

(Remarks on code availability)

We can verify that the SRA uploads of the raw metagenomics data has been completed. I cannot locate the metabolomics data, however.

With respect to code on github, the readme file is inadequate. The readme file is missing key components such as what input files should look like and a list of functions said program can run, and generally how to run their scripts. In other words, it seems the code works, but using it might be a hassle as it does not seem very user friendly. The code itself also seems to not handle errors well from what I could tell, so depending on the dataset used, the program might not work all together (in other words, an error might cause this code to fatally fail). The authors claim that the results files are uploaded on the github repository, but the files need to be reuploaded as they are corrupted. The format is unacceptable to excel even though it is an excel sheet; I tried multiple ways to make sure it wasn't me! Resolution of these details would be needed for publication.

It would be useful for the authors to provide a simple single CSV file of the proportions (and/or read counts) of the individual microbial taxa in each swab sample used in these analyses. This could also be done for the metabolomics data (relative abundance data). This would provide an easily accessible data format such that others could replicate these findings.

We appreciate the excellent feedback we received from the reviewers and the editor. Below are the original comments from the review in **black** and our responses in **blue**.

Reviewer #1 (Remarks to the Author):

Summary: Overall, this manuscript describes an interesting attempt to look at the interactions among different bacterial strains that are commonly found in the healthy and less healthy vaginal biome. Aside for the absence of key components of the manuscript (e.g., figure legends), there are some very interesting outcomes. Thus, the biomass flux analyses are unique and do lead to the conclusions described. Differential clustering of the multiple *Gardnerella* genome species is interesting although the mechanisms are unclear. The co-culturing experiments (e.g., in spent media from different taxa) are interesting in that they may begin to get at the mechanisms of the exclusion and mutualism observed. The mass spec identification of ~85 metabolites was also interesting, and the focus on a couple of the compounds that were found in more than one taxon followed up by testing their effects in vitro was a major plus. A concern of the manuscript is how the taxa were selected. Probably the most important taxon omitted from the study was *Lachnocurva vaginae* (previously BVAB1). That said, the authors may be excused of this since BVAB1 is difficult/impossible to culture as of yet. The absence of *Lactobacillus crispatus* in the analysis was also noteworthy. This taxon is by most measures a hallmark of a healthy vaginal ecosystem. However, since this manuscript was focused on bacterial vaginosis, it is maybe understandable that there was almost no work on *Lactobacillus crispatus*, as *L. crispatus* is almost never associated with BV. Another concern was that despite an extensive Methods section, the methodology for analyzing bacterial biomass seems to be lacking, possibly based on metagenomics, although the details are lacking as are most of the details of the microbiome characterization.

We appreciate the reviewer's thoughtful summary of many of the key aspects of this study and their insight regarding the selection of taxa in our analyses. The omission of *Lachnocurva vaginae* (previously BVAB1) was indeed due to its low co-occurrence with *Gardnerella vaginalis* based on our metagenomic analysis and selection criteria. While we recognize its potential significance in BV, our data did not show it to be highly abundant or frequently co-occurring with *Gardnerella vaginalis*, which therefore limited its inclusion in our study. Future studies may consider different selection criteria or more comprehensive analyses to include this important taxon.

As further discussed below, we have further addressed the critiques from this reviewer including the inclusion of figure captions, methods for analyzing bacterial biomass, among others.

- This reviewer did not have access to the information on the cohort, SOPs for collection of samples, processing of samples, etc. There is a reference to De Seta et al. 2019. It would be helpful to have a brief outline of the cohort, how they were selected, what were inclusion/exclusion criteria, how they were sampled (clinician, self-sample, ?),

pregnant/nonpregnant, where in cycle, etc.

We have included the paragraph below to clarify participant selection and inclusion/exclusion criteria:

“Analysis of vaginal organism co-occurrence was generated using data from participants on the Evvy platform (IRB# 20220118.evvy). Sample collection, processing, shotgun metagenomic sequencing (NovaSeq, Illumina), and bioinformatics analysis was completed as described previously (Thomas-White 2024). Samples were selected based on self-reported symptom profile. *Gardnerella*-dominant profiles were selected and then stratified as either asymptomatic or symptomatic, defined as the presence of self-reported excessive discharge and vulvovaginal pruritus.”

- It is unfortunate that the methodology for assessing the microbiome profiles was not described. Apparently, the analysis was of metagenomic sequences, as was briefly mentioned in the Discussion section of the manuscript. However, no data was presented concerning the sample processing, library generation, sequencing technology, the sequencing strategy (Illumina short read 2 x 150 b PE??), the number of reads per sample, post sequencing data processing, analysis, etc.

In the updated manuscript we have included the citation to Thomas-White 2024, which provides detailed methodology regarding microbiome profile analysis.

- Lines 77-85 and Figure 1. (Note, a significant challenge in reviewing this manuscript was the apparent lack of Figure legends. This reviewer did not have access to them). While the co-occurrence of bacterial taxa is interesting, some measure of absolute abundance might be more relevant. The current analysis is based on relative abundance, but it seems the analysis would be better if based on accurate biomass measurements. However, metagenomics is probably not an ideal strategy to measure in vivo biomass, although with some strategies it might be somewhat informative. That would depend on careful sample collection, which was not described. There is no evidence that this was considered in this study. It is possible to get reasonably accurate measures using qPCR or related technologies. However, even with qPCR, the results vary with the sample that is collected on the swab. Hence, swabs from women with BV may have more total (and more bacterial) biomass than healthy women. It is not clear to this reviewer how this might change the results.

We apologize for the missing figure legends and ensure they are included in the revised manuscript in the section titled “Figure titles & legends”. While we agree that absolute abundance measures would enhance the analysis, our study used relative abundance due to the constraints of metagenomics as highlighted by the reviewer. We are unable to perform qPCR on the samples for accurate biomass measurements so such data is unfortunately outside the scope of the current study. We have highlighted this caveat in our updated discussion section as follows: “While our study provides valuable insights using relative

abundance via metagenomics, it is important to note that this approach may not accurately reflect the absolute bacterial loads, which could be more precisely quantified using qPCR.”

- Paragraph lines 86-90 seem to suggest that this team first identified '13 genetically distinct species'. This is not new. If they found all 13 species in their data sets, that should be made clearer.

Thank you for highlighting the ambiguous language regarding *Gardnerella* speciation, the updated manuscript has included language to clarify that these are not novel species.

- This reviewer finds the heat maps on Figure 2 nearly impossible to interpret. We take it on confidence that the relationships were as described in the manuscript. It's not clear if they heat maps could be presented in a better way. Minimally, it would be helpful to have a figure legend.

We apologize for the missing figure legends and ensure they are included in the revised manuscript in the section titled “Figure titles & legends”, and we have worked to improve figure quality to ensure readability via vectorized images to maintain quality when magnifying the image for interpretation.

- The growth curves in Fig.4 show some interesting interactions between spent media from bacterial cultures and growth rates of other bacteria. In most cases, the effect is negative. Interestingly, *G. vaginalis* seems to have the greatest impact.

We're glad to read the reviewer also made note of these interesting results!

Conclusions. This manuscript represents a significant effort to begin to identify the mechanisms by which bacterial taxa in the female reproductive tract interact positively or negatively. The results seem robust and likely to be reproducible in other cohorts. One common suggestion by reviewers is to test results in an alternate cohort. The major concern here is basing the biomass comparisons on apparently relative abundance readings. Accurate readings of absolute abundance would provide more accurate measures of biomass and could lead to some very significant differences in outcomes.

Given many of the key results (e.g., which metabolites may drive interactions between species) is not dependent on absolute abundance measures, we remain confident in the results, which the reviewer points out as robust. However, we certainly note that additional data like absolute abundance measures would have added value and we have added text to the discussion to highlight this point.

Reviewer #1 (Remarks on code availability):

I'm not a bioinformaticist. My reading of the code would not be informative.

Reviewer #2 (Remarks to the Author): In this study, we conducted an analysis of metagenomic data obtained from human vaginal swabs in order to ascertain the microbial composition of both symptomatic and asymptomatic cases of BV. To validate these predictions, they carried out *in vitro* growth experiments.

The analysis is a little hard for the general reader to follow. Please clarify the following points.

Introduction

- How relevant are *in vitro* growth experiments to *in vivo* dynamics? Please clarify in discussion.

This is an important consideration and we have updated our discussion section to reflect on the difference in *in vitro* vs *in vivo* dynamics:

“Moreover, while our *in vitro* growth experiments provide valuable insights into bacterial metabolic interactions, they may not fully capture the complexity of *in vivo* dynamics, particularly host-microbiome interactions involving immune responses and epithelial cell function. This consideration underscores the need to develop intricate simulations that go beyond pairwise interactions to more accurately reflect *in vivo* community dynamics.”

- Define in introduction- Average nucleotide identity (ANI)

We updated the manuscript to define average nucleotide identity (ANI) at first mention (results section).

- What is GENREs... define it in intro and methods. Discuss relevance in discussion.

We have updated the introduction to include a brief introduction to GENREs. We have also described their use in the discussion:

“GENREs are a powerful tool for predicting metabolic interactions because they allow for the simulation of complex biological systems and the identification of potential metabolic pathways and interactions. By providing a framework to analyze thousands of pair-wise interactions, GENREs help uncover the metabolic dynamics that are not immediately apparent through experimental methods alone.”

- Need legends for all figures

We apologize for the missing figure legends and ensure they are included in the revised manuscript in the section titled “**Figure titles & legends**”.

Results

- Clarify what is “Number of Tests” and “Number of Users” in Table 1.

We have updated Table 1, as reflected below, to improve clarity regarding number of participants versus number of tests.

	Symptomatic	Asymptomatic	Healthy
# Participants	208	468	147
Total number of tests from participants *note some participants collected multiple tests	212	504	154

- How was co-occurrence calculated? (line 251)

Co-occurrence frequency calculation is outlined in the methods section, which is specifically relevant to identifying species that co-occur with *Gardnerella vaginalis* in the context of bacterial vaginosis.

- Define pyani and AN1b analysis. (line 270)

Pyani is defined in the methods section (excerpt below) at first mention, and further information regarding pyani and the AN1b method can be found in the Pritchard et al 2015 and Goris 2007.

“The nucleotide similarity between the Gardnerella genomes was determined using ANI comparison (Goris et al., 2007). 222 Gardnerella genomes were input to pyani, a Python implementation tool for ANI analysis (Pritchard et al., 2015).”

- What is t-SNE on the axis in Fig2 D, E, F. They should be defined in Methods. Explain t-SNE (line 318)

We understand the confusion regarding the lack of axis labels. In the context of t-SNE axes have no global meaning. What this means is that the actual point positioning is not important, only the relationships between the points themselves. We have included an expanded meaning of t-SNE (T-distributed stochastic neighbor embedding), in addition to the citation for the primary literature on the method.

- Need a clearer explanation what is being shown in Figure 2:

We apologize for the missing figure legends and ensure they are included in the revised manuscript in the section titled “**Figure titles & legends**”.

Figure 2 illustrates pairwise *in silico* bacterial interactions through a heatmap and t-SNE plots. The heatmap sections (A, B, C) show the primary bacteria biomass flux change in the presence

of co-occurring bacteria during mutualism, competition, and net interactions, respectively. The t-SNE plots (D, E, F) provide a reduced-dimensional representation of these biomass flux changes across all co-occurring bacteria for mutualism, competition, and net interactions, offering a comprehensive visual summary of the interaction effects.

- Need a clearer explanation of what is being shown in figure 3

We apologize for the missing figure legends and ensure they are included in the revised manuscript in the section titled "Figure titles & legends".

Figure 3 illustrates the balance between relative competition and mutualism among bacteria and the metabolites that underlie these interactions. Panel (A) shows the average biomass increase due to mutualism benefit versus the average biomass decrease due to competition cost for each bacterium. Panel (B) presents a dendrogram of *Gardnerella* species, highlighting the top four most mutualistic and competitive bacteria. Panel (C) identifies the most highly competed for and/or shared metabolites across all interaction simulations, while panel (D) displays *G. piotti* growth curves in NYC III enriched media and media supplemented with 0.1% L-histidine, demonstrating significant growth differences (one-tailed t-test; $t = 3.487$, $p\text{-value} < 0.01$).

- I assume Figure 4 are growth curves... need clarification.

We apologize for the missing figure legends and ensure they are included in the revised manuscript in the section titled "Figure titles & legends".

Figure 4 illustrates pairwise *in vitro* bacterial growth interactions. Panels (A-H) show growth curves in NYC III enriched media (black), spent *G. piotti* media (light pink), and spent *G. vaginalis* media (dark pink/red), with mean and two standard deviation spread. *G. piotti* and *A. christensii* are noted for their highly flocculent phenotypes.

- Clarify the utility of using spent media?

We included the following statement in our discussion section to address the utility of spent media: "By using spent media, we evaluate how bacterial growth is influenced by the metabolic byproducts and nutrient depletion caused by other bacteria, providing a more realistic insight into inter-bacterial dynamics."

- Colors in Figure 5 are unreadable. I assume these are volcano plots?

We apologize for the missing figure legends and ensure they are included in the revised manuscript in the section titled "Figure titles & legends", and we have worked to improve figure quality to ensure readability via vectorized images to maintain quality when magnifying the image for interpretation.

- List Compounds for table 2 in appendix.... They would be of interest to the reader.

All of the mass spectrometry data has been included in full and more clearly referenced in the methods section as ST1 and ST2.

Reviewer #3 (Remarks to the Author): This paper by Dillard et al is a data analysis project of metagenomics and metabolomics datasets from vaginal swab specimens to infer mutualistic and competitive interactions between bacterial species. The paper shows various the statistical and mathematical models used to identify relationships between vaginal bacteria and metabolites, and then tests some of these relationships using in vitro methods and co-culture. The experiments are not clearly outlined in the results from one section to the next, making it difficult to understand the source of the data (in vitro? In vivo?) with the accompanying figures. The figures do not include legends. The data on the study participants is absent in critical areas including how they define BV, which is one of the major variables in these analyses. It is also unclear what is the source of the metagenomic data? (is this published? Did they recruit participants?). Overall, this paper content is difficult to understand, misses critical information, and therefore makes it challenging to comment on the significance of this study.

Specific comments:

- There is no description of the study participant recruitment and consent information, nor the clinical and lab criteria for the definition of BV, which is a critical component of the analysis in this paper. Was this scored by Nugent, Amsel's, etc? How are they defining asymptomatic BV? In the absence of these data it makes it challenging to evaluate the downstream data analysis.

We have updated the methods section to include clarification regarding inclusion criteria:

“Analysis of vaginal organism co-occurrence was generated using data from participants on the Evvy platform (IRB# 20220118.evvy). Sample collection, processing, shotgun metagenomic sequencing (NovaSeq, Illumina), and bioinformatics analysis was completed as described previously (Thomas-White 2024). Samples were selected based on self-reported symptom profile. *Gardnerella*-dominant profiles were selected and then stratified by either asymptomatic or symptomatic, defined as the presence of self-reported excessive discharge and vulvovaginal pruritus.”

- There are no figure legends making it difficult to understand the content of the figures

We apologize for the missing figure legends and ensure they are included in the revised manuscript.

- Table 1 is confusing. What is the data in the first two rows? Participants? What are 'tests'?

Thank you for your feedback, we have updated Table 1, as reflected below, to improve clarity regarding number of participants versus number of tests.

	Symptomatic	Asymptomatic	Healthy
# Participants	208	468	147
Total number of tests from participants *note some participants collected multiple tests	212	504	154

- What is the experiment for Figure 2 in the results section?

We apologize for the lack of clarity due to missing figure legends. Figure 2 represents an *in silico* experiment where we simulate pair-wise metabolic interactions between each possible pair of bacterial strains.

- While I am sure the matrix plots of figure 2 are impressive, what is the data supposed to represent? What is on the y and x axes? It looks like a comparison of *Aerococcus* vs *Aerococcus* based on the color legend.

The coloring of the top row and first column of figure 2 represent the respective species or genus (defined in the Bacteria Classification legend) of either the primary bacteria (y-axis) or the co-occurring bacteria (x-axis). We have included a definition of primary vs co-occurring bacteria in our updated methods section, as stated below:

“In the context of these *in silico* simulations, primary bacteria are the focal bacterial species whose biomass flux changes are measured and analyzed. Co-occurring bacteria are those that interact with the primary bacteria within the simulation environment. Their presence influences the biomass flux of the primary bacteria through competitive or mutualistic interactions, thereby affecting the primary bacteria’s metabolic dynamics and overall community structure. Through our high-throughput pairwise simulations, each bacterial species serves as both the primary and co-occurring bacteria at different points. This means that for each interaction pair, we generate two outputs: one where bacteria 1 is the primary and bacteria 2 is co-occurring, and one where bacteria 1 is co-occurring and bacteria 2 is primary.”

We appreciate the Reviewer considerations of the revised version of our manuscript. Below we provide the original comments in **black** and our responses in **blue**.

REVIEWER COMMENTS

Reviewer #1 (Remarks to the Author):

This is an interesting step toward understanding the interactions among vaginal bacteria and their contributions to bacterial vaginosis (BV). It underlines the likely conclusion that BV is not a single disease entity, but rather is more complex and requires more nuanced definition. Overall, although much more work is required, the data presented support the general conclusions, and the work represents a significant contribution to the field.

The response to the initial reviews is appropriate and comprehensive.

We thank the reviewer for the positive comments.

Reviewer #2 (Remarks to the Author):

The authors have adequately addressed the prior review.

We thank the reviewer for taking the time to review this manuscript.

Reviewer #3 (Remarks to the Author):

While this reviewer asked for this information in the first round, the authors still did not provide sufficient information about the study population, sampling procedures, clinical criteria, etc. In their resubmission they provided only an additional 1-2 sentence description on the clinical cohort which references a pre-print article. This is unacceptable for a journal publication. All of the information on this cohort should be provided in a standalone manner. For example, their definition of BV is flawed. They are representing the self-report of vaginal symptoms as the same as 'BV', which is not necessarily BV (it could be any of a long list of gynecological conditions). It is not clear who gave this information? The participant? The clinician?. The inclusion of a robust description of the clinical population, clinical or laboratory criteria for definition of BV, the collection and storage of biospecimens, the processing of samples, are all an absolute requirement in my opinion. Perhaps the authors should consult with the physician/gynecologist who was the clinical lead in this study to review this information for this publication to ensure its accuracy.

We thank the reviewer for the constructive comments.

The “preprint” reference in the previous version of the manuscript is now a published, peer-reviewed article in *Diagnostics*. We have updated the citation in the manuscript:

Thomas-White, K.; Hilt, E. E.; Olmschenk, G.; Gong, M.; Philips, C.; Jarvis, C.; Sanford, N.; White, J.; Navarro, P. (2024). An Accurate Metagenomics Pipeline to Characterize

We have included a new section in the methods titled “Sample Collection and Cohort Characteristics” which outlines the Evvy platform for requesting an Evvy test, the process of sample collection, sample processing, and the report output. Additionally, we added further details about the cohort in this study including a table (Table 1) that specifies demographic and clinical characteristics of the study population.

We can confirm that all study participants provided informed consent, and study procedures were conducted in accordance with protocols approved by a federally accredited Institutional Review Board (IRB# 20220118.evvy). We have updated the manuscript in the Methods section titled “Sample Collection and Cohort Characteristics” to explicitly state this.

We have made the host depleted raw sequencing data (fastq files) publicly available in the sequence read archive (SRA) under the following BioProject accession number: PRJNA1219227.

The MAGs used in this study were previously uploaded onto public repositories. We have included a supplementary table (S3) that specifies the Genome ID, Genome Name, NCBI Taxon ID, Species, Strain, bioProject Accession, BioSample Accession, Assembly Accession, and GenBank Accession information for each of the publicly available MAGs used in this study. No MAGs were assembled de novo because of this study. The publicly available MAGs were used for determining taxonomic classification of the raw sequencing reads. We have included language in the manuscript to specify this in the Methods section “*Gardnerella* and co-occurring species comparison”.

We have updated our data availability statement to the following: “The metabolomics data generated during this study is available in a public Github repository: https://github.com/lrd3uu/bacterialvaginosis_interactions. All raw metagenomic sequencing data is available on the sequence read archive (SRA) under the following BioProject accession number: PRJNA1219227. All MAGs used in this study for taxonomic classification of metagenomic reads were previously made publicly available and can be accessed via BioProject accession numbers listed in S3”

We acknowledge that self-reported BV symptoms are not the same thing as a BV diagnosis. To address this comment, we have changed the language in our manuscript to define the patient population with *Gardnerella* dominance (according to metagenomic sequencing) and BV-like symptoms as the population with “BV-associated symptoms”, rather than the “BV” population:

“Samples for co-occurrence analysis were selected based on self-reported symptom profile. In the absence of microscopy based diagnostics, Amsel Criteria and Nugent score, BV-associated profiles were defined by *Gardnerella* dominance via metagenomic sequencing.”

REVIEWERS' COMMENTS

Reviewer #3 (Remarks to the Author):

Thank you to the authors for addressing my previous comments. The manuscript is significantly improved with the inclusion on information of the study cohort. This was a major gap in the previous version and having this as a component of the paper greatly improves the quality of the paper.

I have a few minor comments that I think can be quickly addressed with consultation with the editor:

- The BV definition: I appreciate the clarity in how the authors clearly how they are defining 'BV' as "In the absence of microscopy based diagnostics, Amsel Criteria and Nugent score, BV-associated profiles were defined by Gardnerella dominance via metagenomic sequencing (further details in Methods)". However, I would recommend that they use the more modern definition "molecular-BV". This term was adopted after a vigorous discussion by a consortium of scientists and OBGYN in the area of BV research to help refine these terminologies such that we have a consistent nomenclature. The reference for this discussion was in McKinnon et al, "The Evolving Facets of Bacterial Vaginosis: Implications for HIV Transmission", AIDS Res and Human Retroviruses, 2019. doi: 10.1089/AID.2018.0304. This would then remove any ambiguity about whether this is Nugent-BV, Amsel-BV, or molecular-BV.

We have clarified in the manuscript that we are referring to molecular-BV and have cited the above reference.

- Symptoms in Table 1. Could the list of symptoms be listed in the Materials and Methods that are being referenced for BV? I could not find them in the manuscript. Please include them in the Results section where the cohort is first described as well as the methods section.

In the methods section titled "Co-occurrence analysis" we specify that symptomatic is "defined as the presence of self-reported excessive discharge and vulvovaginal pruritus." Additionally, in this section we state that "the symptomatic BV group experienced excessive discharge and itchiness (either internal or external)". We have added this definition in the results section "BV-associated symptom status shapes distinct in vivo co-occurrence patterns among bacterial species" for clarity.

- information of the source data needs some improvement, and we've provided suggestions in the other section.

We have added clarity as described below.

Reviewer #3 (Remarks on code availability):

We can verify that the SRA uploads of the raw metagenomics data has been completed. I cannot locate the metabolomics data, however.

Links to the dropbox folders that house the raw metabolomics data are in the ReadMe file of the following github repository: https://github.com/lrd3uu/bacterialvaginosis_interactions under the “Metabolomics” subheading in the Readme file. We have specified in the “Code availability” section that this repository contains the metabolomics data in addition to the code used in this study.

With respect to code on github, the readme file is inadequate. The readme file is missing key components such as what input files should look like and a list of functions said program can run, and generally how to run their scripts. In other words, it seems the code works, but using it might be a hassle as it does not seem very user friendly. The code itself also seems to not handle errors well from what I could tell, so depending on the dataset used, the program might not work all together (in other words, an error might cause this code to fatally fail). The authors claim that the results files are uploaded on the github repository, but the files need to be reuploaded as they are corrupted. The format is unacceptable to excel even though it is an excel sheet; I tried multiple ways to make sure it wasn't me! Resolution of these details would be needed for publication.

We thank the reviewer for taking the time to review these files and code. We have significantly expanded the readme file to include more details about each of the scripts in each directory. Specifically, we have included a description of the functionality/purpose of all scripts and included the names and formats of input and output files from each of the scripts. We have also uploaded all previously missing input files and output files that relate to each of the scripts. This should allow a user to run the code (with slight modifications to the directory paths to be more user-specific) much more seamlessly.

We also have re-uploaded the previously corrupted excel files in the /Metabolomics directory, which should be downloadable and readable now.

We believe this suggestion greatly improved the github repository and thereby the reproducibility of this computational work.

It would be useful for the authors to provide a simple single CSV file of the proportions (and/or read counts) of the individual microbial taxa in each swab sample used in these analyses. This could also be done for the metabolomics data (relative abundance data). This would provide an easily accessible data format such that others could replicate these findings.

We have provided the relative abundance of individual microbial taxa in each swab used in these analyses. This file is uploaded to the github repository https://github.com/lrd3uu/bacterialvaginosis_interactions under the file name SpeciesRelativeAbundance.csv.